# Alkali basalt from Seifu Seamount in the Japan Sea: post-spreading magmatism in a back-arc setting

Tomoaki MORISHITA[1,2,3], Naoto HIRANO[4], Hirochika SUMINO[5], Hiroshi SATO[6], Tomoyuki SHIBATA[7],
Masako YOSHIKAWA[7], Shoji ARAI[8], Rie NAUCHI[8], Akihiro TAMURA[8]

[1]Faculty of Geosciences and Civil Engineering, Kanazawa University, Kanazawa 920-1192, Japan
[2]Lamont-Doherty Earth Observatory, Columbia University, Palisades, NY 10964, USA
[3]Volcanoes and Earth's Interior Research Center, Japan Agency for Marine-Earth Science and Technology, 2-15 Natsushima, Kanagawa, 237-0061, Japan
[4]Center for Northeast Asian Studies, Tohoku University, 41 Kawauchi, Aoba-ku, Sendai 980-8576, Japan
[5]Graduate School of Arts and Sciences, University of Tokyo, 3-8-1 Komaba, Meguro-ku, Tokyo 153-0041, Japan
[6]Department of Business Administration, Senshu University, 2-1-1, Higashimita, Tama-ku, Kawasaki, Kanagawa 214-8580, Japan
[7]Department of Earth and Planetary Systems Science, Graduate School of Science, Hiroshima University, 1-3-2 Kagamiyama, Higashi-Hiroshima City, Hiroshima 739-8511, Japan
[8]School of Natural System, Kanazawa University, Kanazawa 920-1192, Japan

*Correspondence to*: Tomoaki Morishita (tomo_make_a_wish@icloud.com)

**Abstract.** We present geochemical and $^{40}$Ar/$^{39}$Ar age data for a peridotite xenolith-bearing basalt dredged from Seifu Seamount (SSM basalt) in the northeast Tsushima Basin, southwest Japan Sea. A $^{40}$Ar/$^{39}$Ar plateau age of 8.33 ± 0.15 Ma (2σ) was obtained for the SSM basalt, indicating that it erupted shortly after the termination of back-arc spreading in the Japan Sea. The SSM basalt is a high-K to shoshonitic alkali basalt that is characterized by light rare earth element enrichment. The trace element features of the basalt are similar to those of ocean island basalt, although the Yb content is much higher, indicating formation by low-degree partial melting of spinel peridotite. The Nd, Sr, and Pb isotopic compositions of the SSM basalt differ from those of back-arc basin basalts in the Japan Sea. The Sr–Nd isotopic composition of the SSM basalt suggests its source was depleted mid-ocean ridge mantle containing an EM1-like component. The SSM basalt was formed in a post-back-arc extension setting by low-degree partial melting of upwelling asthenosphere that had previously been associated with the main phase of back-arc magmatism.

Keywords: Back-arc magmatism, Alkali basalt, Japan Sea, Post-back-arc magmatism

## 1. Introduction

Numerous studies have investigated the magmatism that occurs during back-arc rifting (e.g., Martinez *et al.*, 2001; Pearce and Stern, 2006). The Japan Sea is a typical inactive back-arc basin developed between an island arc and a continent (Tamaki and Honza, 1985; Uyeda and Kanamori, 1979), and is located in the northern part of the western Pacific where

back-arc basins are common. Numerous geophysical surveys have been carried out to elucidate the architecture of the Japan Sea back-arc basin and its formation processes (e.g., Lee *et al*., 1999; Yoon *et al*., 2014; Sato *et al*., 2014). Geophysical data and sampling by the Ocean Drilling Program (ODP) have revealed the mechanisms of formation of the northern Japan Sea (Tamaki *et al*., 1992) (Fig. 1). Lithospheric break-up and oceanic spreading began at the eastern margin of the continent at 32–28 Ma, and back-arc magmatism ceased at 18–15 Ma (Tamaki *et al*., 1992). The basins in this region comprise mainly extended continental and oceanic crust (Tamaki *et al.*, 1992). A chain of seamounts and islands are distributed in these basins (Fig. 1). Volcanic rocks from seamounts in the Yamato Basin have ages of 13–6 Ma (Kaneoka *et al*., 1990; Kaneoka and Yuasa, 1988), whereas volcanism on Ulleung Island in the Tsushima Basin (Ulleung Basin) is much younger (<2 Ma; Kim *et al*., 1999).

   One basaltic sample containing mantle xenoliths was dredged from Seifu Seamount (herein called the SSM basalt) in the northeast Tsushima Basin in the southwest Japan Sea (Fig. 1) (Ninomiya *et al*., 2007). Peridotites from back-arc basins have rarely been reported (Ichiyama *et al*., 2013; Ohara, 2006). Such xenoliths could provide insights into the architecture and history of the Japan Sea basins. To evaluate the significance of these xenoliths, we report petrological, geochemical, and geochronological data for this mantle xenolith-bearing alkali basalt sample that was first described by Ninomiya *et al*. (2007).

   Pouclet *et al*. (1995) summarized the magmatic history of the circum-Japan Sea region, including the southwest Japan arc, northeast China, and back-arc basins (Japan Basin, Ulleung Basin, and Yamato Basin). Alkali basalt magmatism became common in the southwest Japan arc after the termination of extension in the Japan Sea at <4 Ma. Kim and Yoon (2017) suggested that seamounts in the Tsushima Basin were formed post-back-arc spreading, based on geochemical data for the alkali basalt samples studied by Lee *et al*. (2011). Sato *et al.* (2002) proposed that post-back-arc spreading magmatism during the last stages of back-arc basin spreading and after the cessation of spreading is a common process in the latter stages of back-arc development. Ishizuka *et al.* (2009) also investigated post-back-arc spreading magmatism in the Shikoku Basin of the Izu–Bonin arc–back-arc system. In this paper, we discuss the origin of the SSM basalt in the context of post-back-arc magmatism.

## 2. Geological setting and sample description

   Seifu Seamount is located at the northeastern margin of Tsushima Basin, which is a junction of three basins that are located between continental crustal fragments (Fig. 1). Tsushima Basin is now geomorphologically connected to the southwestern margin of the Japan Basin by the Ulleung Interplain Gap (UIG), and is separated from Yamato Basin by the Yamato Rise and Oki Bank. The studied basalt sample was dredged from Seifu Seamount by the KT85-15 cruise of *R/V Tansei-maru* of the Ocean Research Institute, University of Tokyo, in 1985 (Dredge Station No. KT85-15 D-3: 38°12.20'–12.80'N and 132°34.70'E) (Shimamura *et al*., 1987). Lee *et al*. (1999) proposed that the basement of Tsushima Basin is thick oceanic crust formed by incomplete extension of continental crust, which was caused by the westward propagation of

extension during opening of the Japan Sea. Tsushima Basin was opened in a N–S direction, as evidenced by the NE–SW orientation of ridge-like features parallel to the Yamato Basin spreading axis (Lee *et al*., 1999). Kim *et al*. (2011) showed that a chain of seamounts buried by sediments occurs along the UIG, and proposed that the ages of the UIG seamounts are the same as those of the Yamato Seamount Chain. Seifu Seamount is located in the Japan Basin in the most ENE region of the UIG seamount chain.

The basalt sample contains peridotite xenoliths of variable size from a few millimeters to <3 cm, although rare xenoliths are up to 10 cm in size, along with associated xenocrysts (Ninomiya *et al*., 2007) (Fig. 2). Ninomiya *et al*. (2007) divided the peridotite xenoliths into two types based on mineral chemistry (Types I and II). Type I xenoliths were interpreted to be residues after relatively high degrees of open system melting caused by light rare earth element (LREE)-enriched fluids–melts, whereas Type II xenoliths have characteristics similar to subcontinental lithospheric mantle.

The basalt has a porphyritic texture, and the phenocrysts are mainly olivine, with small amounts of plagioclase, orthopyroxene, clinopyroxene, and spinel. Plagioclase phenocrysts are anhedral with albite twinning, oscillatory zoning, and dusty zones, and are sometimes surrounded by tiny plagioclase crystals that are the same size as crystals in the groundmass. Orthopyroxene phenocrysts are rimmed by fine-grained mineral aggregates, which are similar to the orthopyroxenes in the peridotite xenoliths, indicating the orthopyroxene phenocrysts are xenocrysts. Spinel phenocrysts are up to 0.5 mm in size, anhedral, and rounded. The groundmass is crystalline, has an intersertal texture, and consists mainly of euhedral plagioclase, with small amounts of olivine that is partly serpentinized–altered, opaque minerals (ilmenite and titanomagnetite), apatite, and clinopyroxene (Fig. 2). Locally, the plagioclase crystals in the groundmass exhibit a weakly developed trachytic texture.

## 3. Geochemistry and dating

### 3.1. Analytical methods

Obvious crustal and mantle xenolith fragments were removed from the basalt sample before crushing to powder. Whole-rock major and trace element compositions of the SSM basalt were determined by electron probe microanalysis (EPMA; JEOL JXA-8800 Superprobe) and laser ablation–inductively coupled plasma–mass spectrometry (LA–ICP–MS) (Agilent 7500S coupled to a MicroLas GeoLas Q-Plus: Ishida *et al*., 2004) at Kanazawa University, Kanazawa, Japan. For the EMPA and LA–ICP–MS measurements, glass was prepared from the powdered sample with an Ir strip heater (Nicholls, 1974; Stoll *et al*., 2008), which comprises Cu electrodes connected to a direct current power supply and an Ir strip. Approximately 20 mg of the whole-rock powder was fused on the Ir strip. A current of 42–45 A and heating duration of <30 s were used to make the glass. The fused melt was quenched in an air stream after disconnecting the power supply. No visible residual and quench phases were observed in the glass. Details of this analytical method are described by Tamura *et al*. (2015).

EMPA was conducted with an accelerating voltage of 15 kV, beam current of 20 nA, and beam diameter of 10 μm. Natural and synthetic mineral standards provided by JEOL were used for all analyses. JEOL software using ZAF corrections

was used for data reduction. X-ray peaks of all elements were counted for 10 s. Sodium and K intensities were monitored during analysis, and no loss of these elements was detected. BHVO-2 glass from the United States Geological Survey (Reference Materials Project; USGS) was analyzed repeatedly to assess data quality. Data precision was better than ±5% and ±20% (relative standard deviation) for elements with abundances of >0.5 and <0.5 wt.%, respectively (Table 1).

5           NIST SRM 612 standard glass was used as the primary calibration standard during LA–ICP–MS analysis, and was analyzed at the beginning of each batch of ≤8 unknowns. A linear drift correction was applied between each calibration standard analysis. The elemental concentrations of NIST SRM 612 are the preferred values of Pearce *et al*. (1997). Data were also corrected using $^{29}Si$ as an internal standard, based on the Si contents obtained by EMPA, following the protocols of Longerich *et al*. (1996). NIST SRM 614 and BHVO-2 glasses were analyzed for data quality control purposes. The

measured concentrations for NIST SRM 614 and BHVO-2 are within 10% of previously reported data for these standards, and the precision was better than ±10% (relative standard deviation) for all the analyzed elements.

           The major element composition was also determined by X-ray fluorescence spectrometry (XRF; Rigaku Supermini) at Senshu University, Tokyo, Japan. The powdered sample was dried at 950°C and a mixture of this and lithium tetraborate ($Li_2B_4O_7$) flux (1:5 weight ratio) was melted at 1200°C. The analysis was performed using an accelerating voltage of 50 kV

and beam current of 4.0 mA. Details of the XRF methodology are given by Sato (2010).

           The analytical procedures used for chemical separation and isotopic analysis of Sr, Nd, and Pb are outlined by Yoshikawa and Nakamura (1993), Shibata and Yoshikawa (2004), and Miyazaki *et al*. (2003), respectively. Mass spectrometry was conducted with a Thermo-Finnigan MAT262 equipped with nine Faraday cups in static multi-collection mode. Normalizing ratios used to correct for isotopic fractionation were $^{86}Sr/^{88}Sr = 0.1194$, $^{146}Nd/^{144}Nd = 0.7219$, and 0.061

per atomic mass unit for Pb isotopes. Measured isotopic ratios for standard materials were $^{87}Sr/^{86}Sr = 0.710261 \pm 0.000018$ (2 sigma) for NIST 987, $^{143}Nd/^{144}Nd = 0.511842 \pm 0.000018$ (2 sigma) for La Jolla, and $^{206}Pb/^{204}Pb = 16.937 \pm 0.006$ (2 sigma), $^{207}Pb/^{204}Pb = 15.491 \pm 0.0048$ (2 sigma), and $^{208}Pb/^{204}Pb = 36.721 \pm 0.026$ (2 sigma) for NIST 981. Total procedural blanks for Sr, Nd, and Pb were <100, <10, and <10 pg, respectively. The geochemical and isotopic data for the SSM basalt are summarized in Table 1.

We undertook $^{40}Ar/^{39}Ar$ dating of the SSM basalt. The sample was crushed into grains that were < 2 millimeters in size, from which fresh groundmass was separated. Samples were wrapped in Al foil and loaded into an Al capsule (70 mm in length and 10 mm in diameter) with flux monitors EB-1 biotite (91.4 ± 0.5 Ma; Iwata, 1997), $K_2SO_4$, and $CaF_2$. The samples were irradiated for 24 h in the Japan Material Testing Reactor (JMTR). During irradiation, the samples were shielded by Cd foil in order to reduce thermal neutron-induced $^{40}Ar$ production from $^{40}K$ (Saito, 1994). The Ar extraction and isotopic

analyses were conducted at the Radioisotope Center, University of Tokyo, Tokyo, Japan. During step heating, the gas was extracted in eight steps between 600 and 1300°C. The dating method has been described by Ebisawa *et al*. (2004).

### 3.2. Results

           $SiO_2$, $TiO_2$, and total alkali contents ($Na_2O + K_2O$) of the SSM basalt are 49, 1.9, and 5.8 wt.% ($Na_2O = 4.0$ wt.%

and $K_2O$ = 1.8 wt.%), respectively. The SSM basalt is classified as an alkali basalt in a total alkalis vs. $SiO_2$ diagram (Miyashiro, 1978) (Fig. 3) and low-Ti basalt ($TiO_2$ < 2.5 wt.%, Ti/Y < 500: Xu et al., 2001). The MgO content and FeO*/MgO ratio (FeO* = total FeO) are 6.4 wt.% and 1.5, respectively.

A primitive-mantle-normalized trace element pattern is shown in Fig. 4. The SSM basalt is characterized by LREE enrichment and no apparent anomalies in Eu and high-field-strength elements (HFSE) relative to neighboring REE, whereas Rb and Th exhibit slight negative anomalies. $(Nb/La)_{PM}$ and $(Ta/La)_{PM}$ ratios are both ca. 1.5 (Fig. 5a), and $La_{PM}$ and $(La/Yb)_{PM}$ ratios are 54 and 9, respectively. The trace element pattern of the SSM basalt is similar to those of ocean island basalts (OIB). However, the $Yb_{PM}$ content (6) is distinctly higher than those of OIB (Fig. 5b).

Isotopic analyses of the unleached and leached sample (6M HCl) were undertaken to evaluate the effects of alteration (Table 2). Although these isotopic compositions are slightly different, alteration has not significantly affected the isotopic data. $^{143}Nd/^{144}Nd$ and $^{87}Sr/^{86}Sr$ ratios of the leached sample are 0.512903 and 0.703476, respectively (Fig. 6). The Pb isotopic ratios of the SSM basalt are $^{206}Pb/^{204}Pb$ = 17.664, $^{207}Pb/^{204}Pb$ = 15.434, and $^{208}Pb/^{204}Pb$ = 37.308 (Fig. 6).

The $^{40}Ar/^{39}Ar$ age spectrum of the SSM basalt has a plateau age of 8.33 ± 0.15 Ma (2σ) in six fractions at lower temperatures (Fig. 7 and Table 3), with an initial $^{40}Ar/^{36}Ar$ ratio obtained from an inverse isochron that corresponds to the atmospheric ratio (295.5; Fig. 7).

## 4. Discussion

### 4.1. Timing of SSM basaltic magmatism

Basalt samples were recovered from the Japan Sea by the ODP Leg 127/128 cruises in the Yamato Basin (Sites 794 and 797) and Japan Basin (Site 795) (e.g., Tamaki *et al*., 1992) (Fig. 1). $^{40}Ar/^{39}Ar$ dating of these basalts indicated formation during back-arc magmatism at 21–18 Ma (Yamato Basin) and 25–15 Ma (Japan Basin) (Kaneoka *et al*., 1992). Back-arc basin basaltic (BABB) magmatism caused the extension of continental crust until 15 Ma (e.g., Kaneoka *et al*., 1992; Tamaki *et al*., 1992). $^{40}Ar/^{39}Ar$ dating of andesitic rocks from the Yamato Basin seamounts yielded ages of 17–11 Ma (Kaneoka *et al*., 1990). The age of the SSM basalt (8.3 Ma) indicates formation shortly after termination of Japan Sea opening, and is clearly older than volcanism on Ulleung and Jejudo islands (<2 Ma).

### 4.2. Origin of the SSM basaltic melt

The trace element pattern of the SSM alkali basalt shows no depletion of Nb and Ta relative to LREE (e.g., $Nb_{PM}/La_{PM}$ = 1.5) and high heavy REE (HREE) abundances (e.g., $Yb_{PM}$ = 6) (Fig. 5). The pattern is broadly similar to OIB-type alkali basalts from oceanic islands (e.g., Ulleung Island) and continental regions of east China, whereas the high HREE abundances distinguish the SSM basalt from other alkali basalts in the circum-Japan Sea area (Fig. 5). Alkali basalts with

high Nb/La ratios and HREE abundances have also been reported from the South Korean Plateau, located to the east of SSM (Lee *et al*., 2011) (Fig. 5c).

The FeO/MnO ratio of a melt is a proxy for contributions from pyroxenite components in the mantle source (Herzberg, 2011). The FeO/MnO ratio of the SSM basalt is 47, which is consistent with partial melting of peridotite (50–60). Relatively high HREE abundances in the SSM basalt can be explained by the residual phases in the mantle source. Key elemental ratios, such as plots of Sm/Yb vs. Nb/Yb and Zr/Y vs. Nb/Yb, (Fig. 8), combined with simple partial melting models, show that garnet was not a residual phase in the mantle source. Formation of the SSM basalt can be explained by low-degree melting of spinel peridotite.

Geochemical contributions from slab-derived components were evaluated by ratios of elements partitioned into aqueous fluids (i.e., Ba) to elements partitioned into silicate melts (i.e., Th, Nb, and Ta), which were normalized to a compatible element in the subducted slab component (i.e., Yb) (Fig. 9). Using this approach, the SSM basalt has a MORB–OIB composition, whereas Japan Sea BABB magmatism and basalts from southwestern Japan show the addition of slab-derived components.

Due to the very minor contributions from slab components to the SSM basalt source, the SSM basalt can be used to reveal the mantle components beneath the Japan Sea back-arc region. In isotopic plots, the SSM basalt also has a MORB–OIB composition, whereas Japan Sea BABB magmatism is characterized by MORB to island arc tholeiite–ocean island tholeiite (E-type) compositions (Allan and Gorton, 2006; Hirahara *et al*., 2015) (Fig. 7). The Nd–Sr–Pb isotopic compositions of the SSM basalt are depleted and are more similar to the depleted (D)-type Japan Sea BABB magmatism than the other circum-Japan Sea alkali basalts (Fig. 7). Hirahara *et al*. (2015) suggested that Japan Sea BABB magmatism has a composition between DMM and slab-derived components. The D-type basalts were formed from DMM with little or no contribution from slab-derived components. Forward modeling indicated that the melting conditions for the D-type basalts were deeper and hotter than those for MORB (Hirahara *et al*., 2015). It is concluded that the SSM basalt mantle source was likely depleted MORB mantle (DMM) mixed with an enriched mantle component with EM1-like characteristics (Figs. 7 and 10).

### 4.3. Tectonic implications of the SSM basalt

The age and geochemistry of the SSM basalt suggest that upwelling of the Japan Sea back-arc asthenosphere continued after the cessation of back-arc spreading, resulting in a low-degree partial melting (Fig. 10). Sato *et al*. (2002) described post-spreading magmatism that formed seamounts, which comprise enriched basalts, in the back-arc region of Shikoku Basin that formed during and/or after the last stages of back-arc spreading. Sato *et al*. (2002) suggested that post-spreading magmatism is a common process in back-arc basins in the western Pacific. For example, in contrast to the spreading-related magmatism (30–15 Ma), enriched basaltic magmatism (Kinan Seamount Chain; Fig. 1: 15–7 Ma) has been identified in the Shikoku Basin in the Izu–Bonin back-arc (Sato *et al*., 2002; Ishizuka *et al*., 2009). The age of the SSM

basalt (8.3 Ma) is ca. 7 Myr younger than the termination of Japan Sea spreading (~15 Ma), and similar to that of post-spreading magmatism in Shikoku Basin. Ishizuka *et al.* (2009) suggested that post-back-arc spreading magmatism in Shikoku Basin was caused by heterogeneities in upwelling asthenospheric mantle, which produces BABB melts beneath the back-arc spreading center.

5          Alkali basalts from the South Korean Plateau also have distinctive Nd–Sr isotopic compositions, which are similar to the SSM basalt (Fig. 5a), whereas their Pb isotopic compositions are similar to Japan Sea BABB magmatism (Fig. 7). The isotopic compositions of the SSM basalt are similar to those of 13–6 Ma trachyandesites from seamounts in Yamato Basin (Tatsumoto and Nakamura, 1991). Based on geochemical data for South Korean Plateau basalts and the tectonic history of Tsushima Basin, Kim and Yoon (2017) concluded that these basalts were the product of post-spreading magmatism.

However, the age of these basalts is unknown. Numerous submarine volcanoes, including seamounts buried by sediments, were discovered in the Japan Sea (e.g., Kimura *et al.*, 1987; Kim *et al.*, 2011) (Fig. 1). These submarine volcanoes and seamounts were also likely formed after back-arc spreading ceased.

          Based on our study of the SSM basalt coupled with earlier preliminarily works on peridotite xenoliths in the SSM basalt (Ninomiya et al., 2007) suggest peridotite xenoliths in the SSM basalt seem to be not directly related to the SSM

basalt petrogenesis (Fig. 10). One of the two types of peridotite xenoliths is interpreted as fragments of subcontinental lithospheric mantle that were already located before the Japan Sea opening. The other type, which is characterized by residue after open system melting caused by infiltration of LREEs-enriched fluids, may be related to back-arc spreading magmatisms affected by slab-derived fluids/melts (Fig. 10). Further study of peridotite xenoliths and mafic xenoliths is needed to reconstruct the crust-mantle evolution beneath the Japan Sea.

20          Alkali basalts have also been reported from the Philippine Sea Plate (Kinan Seamount Chain; Sato *et al.*, 2002; Ishizuka *et al.*, 2009) and Pacific Plate (petit-spot magmas; Fig. 1; Hirano *et al.*, 2006; Machida *et al.*, 2009) near the Japan Sea in the western Pacific. The geochemical characteristics of the SSM basalt are similar to those of the Kinan Seamount Chain, but different from those of petit-spot magmas (Figs 3–6). The involvement of an EM1 component in the mantle source is a minor but pervasive feature in both the Japan Sea and Shikoku Basin in the western Pacific.

**5. Conclusions**

          The SSM basalt (8.3 Ma) formed ca. 7 Myr after termination of Japan Sea spreading (~15 Ma). The SSM basalt formed by low-degree partial melting of spinel peridotite, and was the result of post-back-arc spreading magmatism in the Japan Sea. The SSM basalt mantle source is consistent with being DMM with a minor EM1-like contribution. Geochemical similarities between post-back-arc magmatism in the Japan Sea and Shikoku Basin of the Philippine Sea suggest the EM1

component is a minor, but widespread, component beneath the western Pacific. Peridotite xenoliths in the SSM basalt do not appear to be directly related to the SSM basalt petrogenesis but will shed light on the crust-mantle evolution beneath the Japan Sea.

**Acknowledgements**

We are grateful to Captain K. Miki, T. Kato, and the other crew members of the *R/V Wakashio Maru*, Captain H. Igarashi and crew members of the *R/V Tansei Maru*, and the scientists that participated on these cruises. Comments from two anonymous reviewers and editor have greatly improved this paper. We also thank T. Ishii for providing the studied sample.

This work was partly supported by Kanazawa University SAKIGAKE 2018 (TM) and Grants-in-Aid for Scientific Research from the Ministry of Education, Culture, Sports, Science, and Technology of Japan (Nos 16H05741 and 19H01990; TM).

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

**Figure Captions**

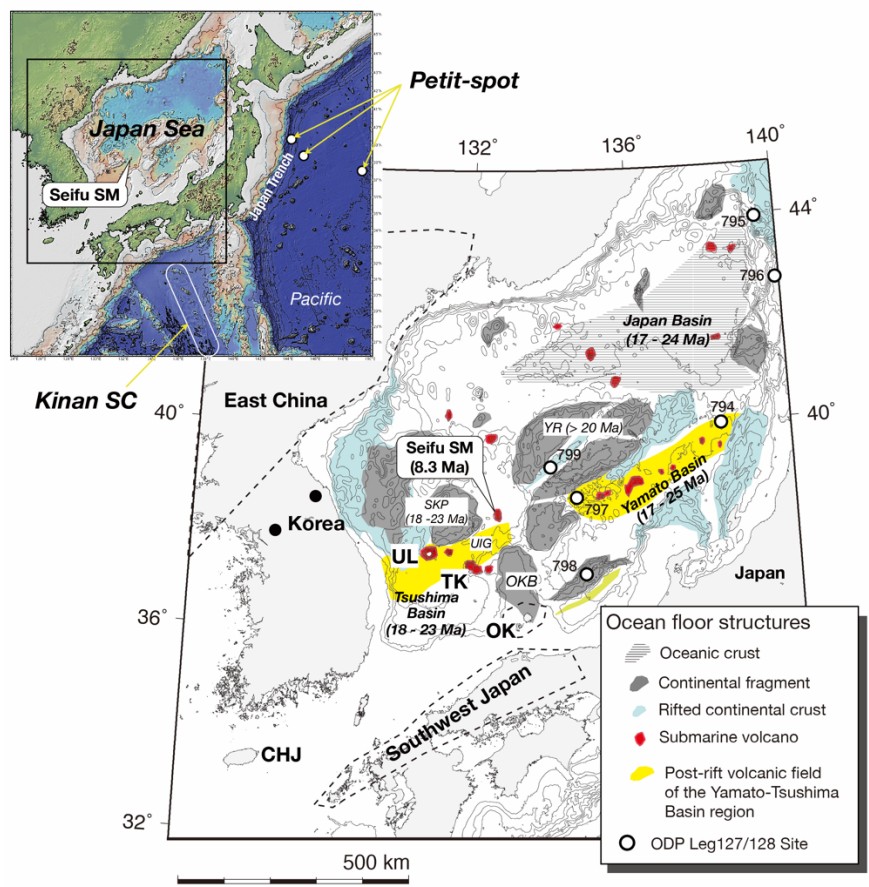

Figure 1: Location of Seifu Seamount (SSM) and ocean floor structure (YR = Yamato Rise; OKB = Oki Bank; SKP = South Korea Plateau; UIG = Ulleung Interplain Gap) in the Japan Sea (modified after Tamaki, 1988; Tamaki *et al*., 1992; Kim *et al*., 2011). Oceanic crust is only found in the Japan Basin. Note that Seifu Seamount is on extended continental crust (white area) in the Japan Basin, located between the seamounts (submarine volcanoes) of the UIG and Yamato Basin (Kim *et al*., 2011). The ages of submarine volcanoes in this area are 17–10 Ma. Ulleung Island (UL), Takeshima Island (TK), Oki Island (OK), and Cheju Island (CHJ) are mainly Quaternary volcanoes (<5 Ma). Sites from ODP Legs 127 and 128 that recovered back-arc basin basaltic (BABB) rocks from Yamato Basin (Tamaki *et al*., 1992) and other locations of basalts in the circum-Japan Sea area are shown (southwest Japan, including San-in and San-yo, northwest Kyushu, Korea, and east China). Petit-spot and the Kinan Seamount Chain are locations of alkali basalts on the Pacific and Philippine Sea plates, respectively (Hirano *et al*., 2006; Ishizuka *et al*., 2009). Age data are from Tamaki *et al*. (1992), Kaneoka *et al*. (1990, 1992), and Kim *et al*. (2011). The topographic and bathymetric maps were prepared using GeoMapApp (Ryan *et al*., 2009).

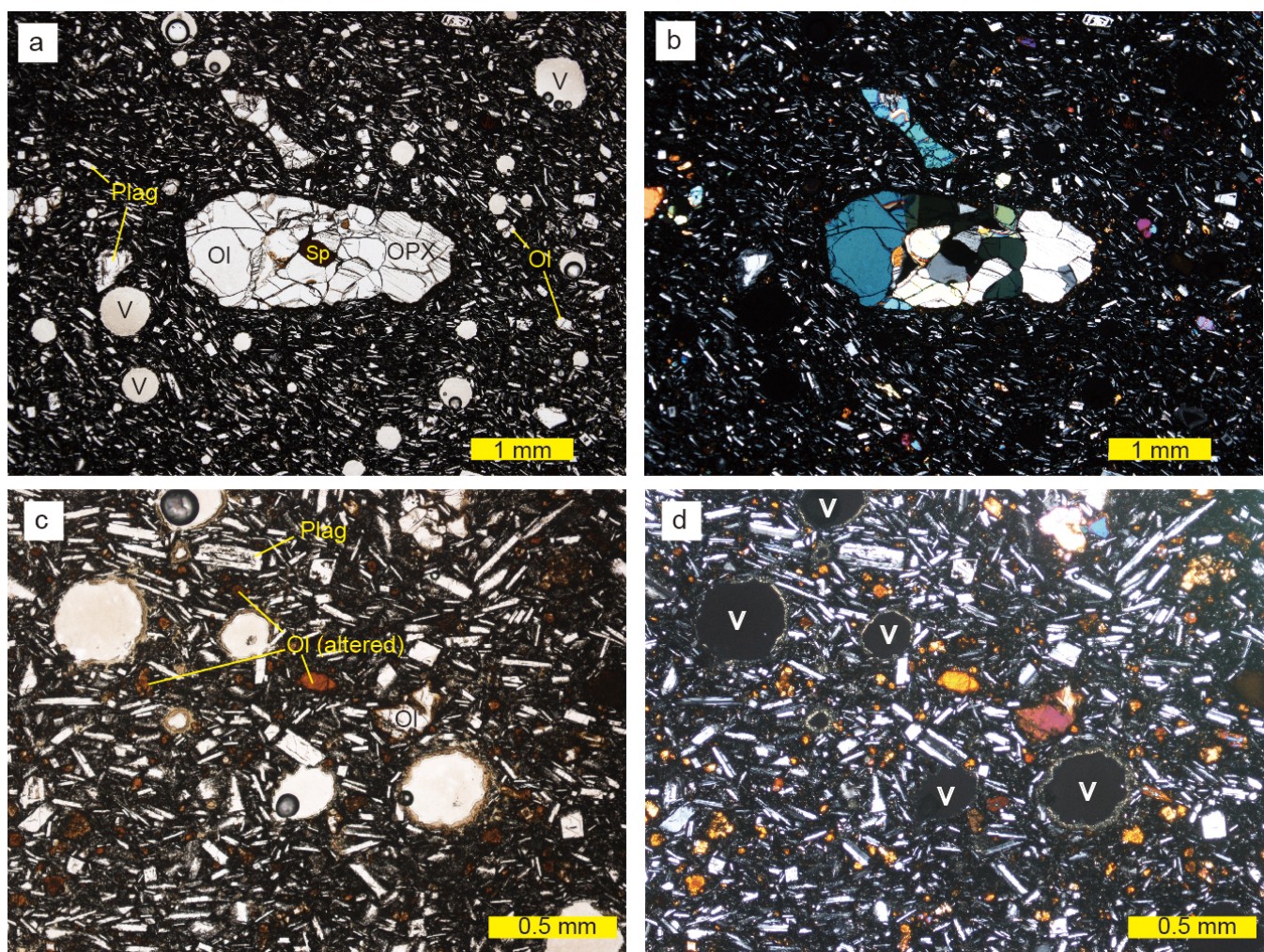

Figure 2: Photomicrographs of the studied Seifu Seamount basalt sample. (a) A peridotite xenolith in plane-polarized light. (b) Cross-polarized light image of (a). (c) Typical texture of the studied Seifu Seamount basalt in plane-polarized light. (d) Cross-polarized light image of (c). Ol = olivine, Plag = plagioclase, Sp = spinel, and V = vesicle.

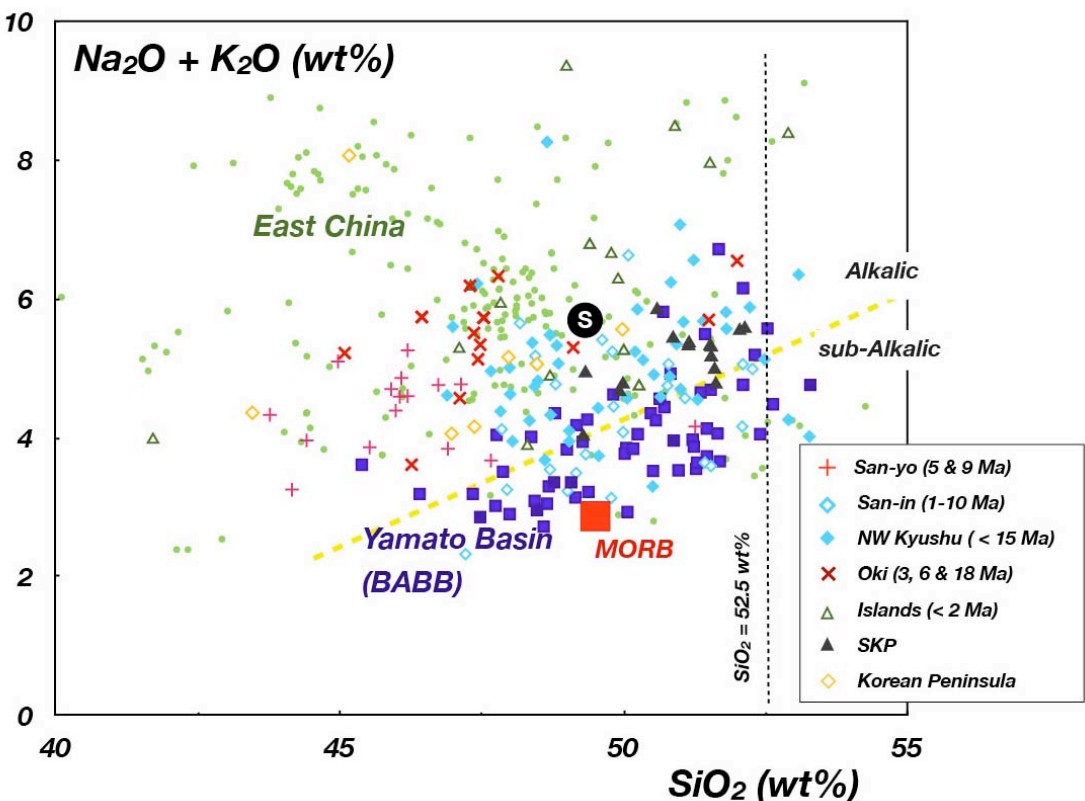

Figure 3: Whole-rock $SiO_2$ and total alkali ($Na_2O + K_2O$) contents of the basalt from Seifu Seamount (black circle annotated with S). The alkali and sub-alkali boundary lines are from Miyashiro (1978). Basalts from the circum-Japan Sea area are shown for comparison. BABB = back-arc basin basalt from the Yamato and Japan Basins. Data sources: Dostal *et al.* (1988), Nakamura *et al.* (1989, 1990), Allan and Gorton (1992), Iwamori (1992), Miyake (1994), Uto *et al.* (1994, 2004), Basu *et al.* (1991), Chung (1999), Kim *et al.* (1999), Pouclet *et al.* (1995), Zou *et al.* (2000), Zhang *et al.* (2002), Choi *et al.*, (2006), Yan and Zhao (2008), Lee *et al.* (2011), and Hirahara *et al.* (2015).

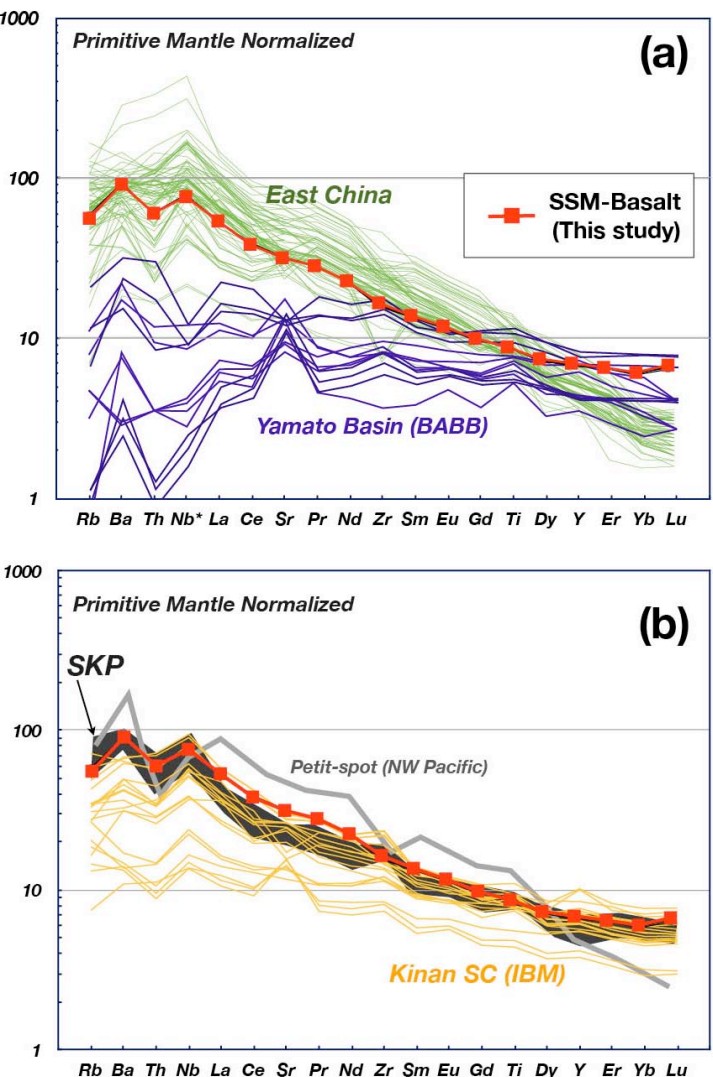

Figure 4: Primitive-mantle-normalized trace element pattern of the basalt from Seifu Seamount (SSM basalt). Normalizing values (primitive mantle) are from Sun and McDonough (1989). Representative basalts from the circum-Japan Sea area are shown for comparison (see Fig. 2 for the data sources). Back-arc basin basaltic (BABB) magmatism from Yamato Basin and alkali basalts from east China are shown in (a). Alkali basalts from the South Korean Plateau (SKP) (black field) are shown in (b). Data for the Kinan Seamount Chain in Shikoku Basin (Izu–Bonin–Mariana region) and petit-spot volcanoes in the western Pacific are from Ishizuka *et al*. (2009) and Hirano *et al*. (2006), respectively. In (a), Nb* was calculated by assuming that $Nb_{PM}/Ta_{PM} = 1$.

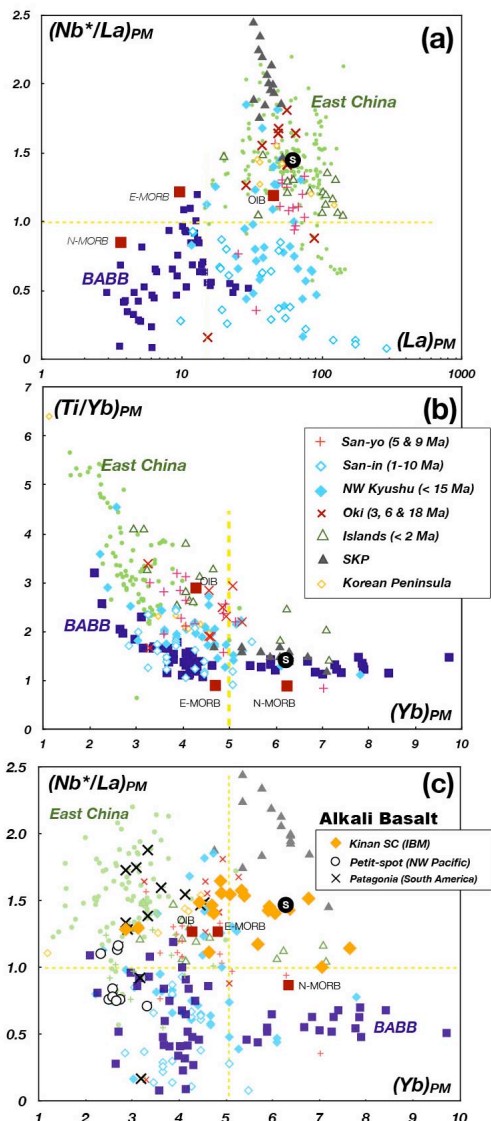

Figure 5: High-field-strength and rare earth element systematics of the SSM basalt (black circle annotated with S) and basalts from the circum-Japan Sea area with $SiO_2 < 52.5$ wt.% (Ulleung, Jejudo, and Takeshima islands; SKP = South Korean Plateau). Normalizing values (primitive mantle) and reference basalt data (MORB and OIB) were taken from Sun and McDonough (1989). (a) Nb vs. La (Nb* = Nb calculated by assuming that $Nb_{PM}/Ta_{PM} = 1$). (b) Ti vs. Yb (data sources are given in Fig. 3). (c) Discrimination of basalts based on the compositional relationships shown in (a) and (b). Note that high-Nb/La and high-Yb basalts have rarely been reported from the circum-Japan Sea area. Data for alkali basalts from the Kinan Seamount Chain in the Shikoku Basin (Izu–Bonin–Mariana region) (Ishizuka *et al*., 2009), petit-spot volcanoes in the western Pacific (Hirano *et al*., 2006), and Patagonia (South America) (Stern *et al*., 1990) are shown for comparison.

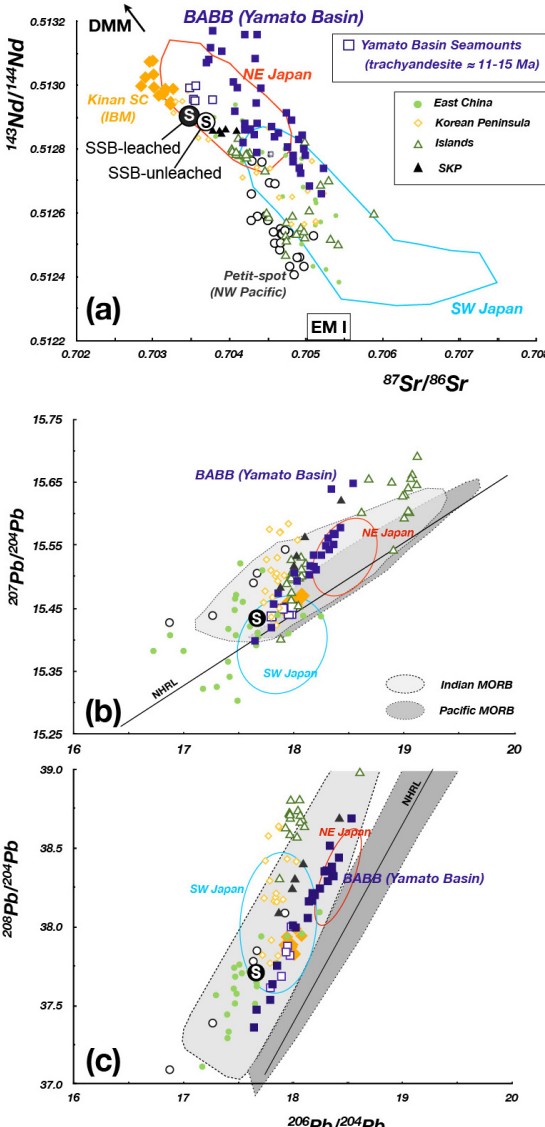

Figure 6: Nd–Sr–Pb isotopic compositions of the SSM basalt (leached: black circle annotated with S; unleached: white circle annotated with S). (a) Sr–Nd isotopic plot. (b) $^{206}Pb/^{204}Pb$ vs. $^{207}Pb/^{204}Pb$. (c) $^{206}Pb/^{204}Pb$ vs. $^{208}Pb/^{204}Pb$. Reference data and data fields were taken from Basu *et al*. (1991), Tatsumoto and Nakamura (1991), Cousens and Allan (1992), Nohda *et al*. (1992), Pouclet *et al*. (1994), Lee *et al*. (2001), Park *et al*. (2005), Choi *et al*. (2006), Ishizuka *et al*. (2009), Machida *et al*. (2009, 2015), and Hirahara *et al.* (2015). Indian and Pacific MORB fields in (b) and (c) are from Miyazaki *et al.* (2015). The Northern Hemisphere Reference Line (NHRL) is from Hart (1984).

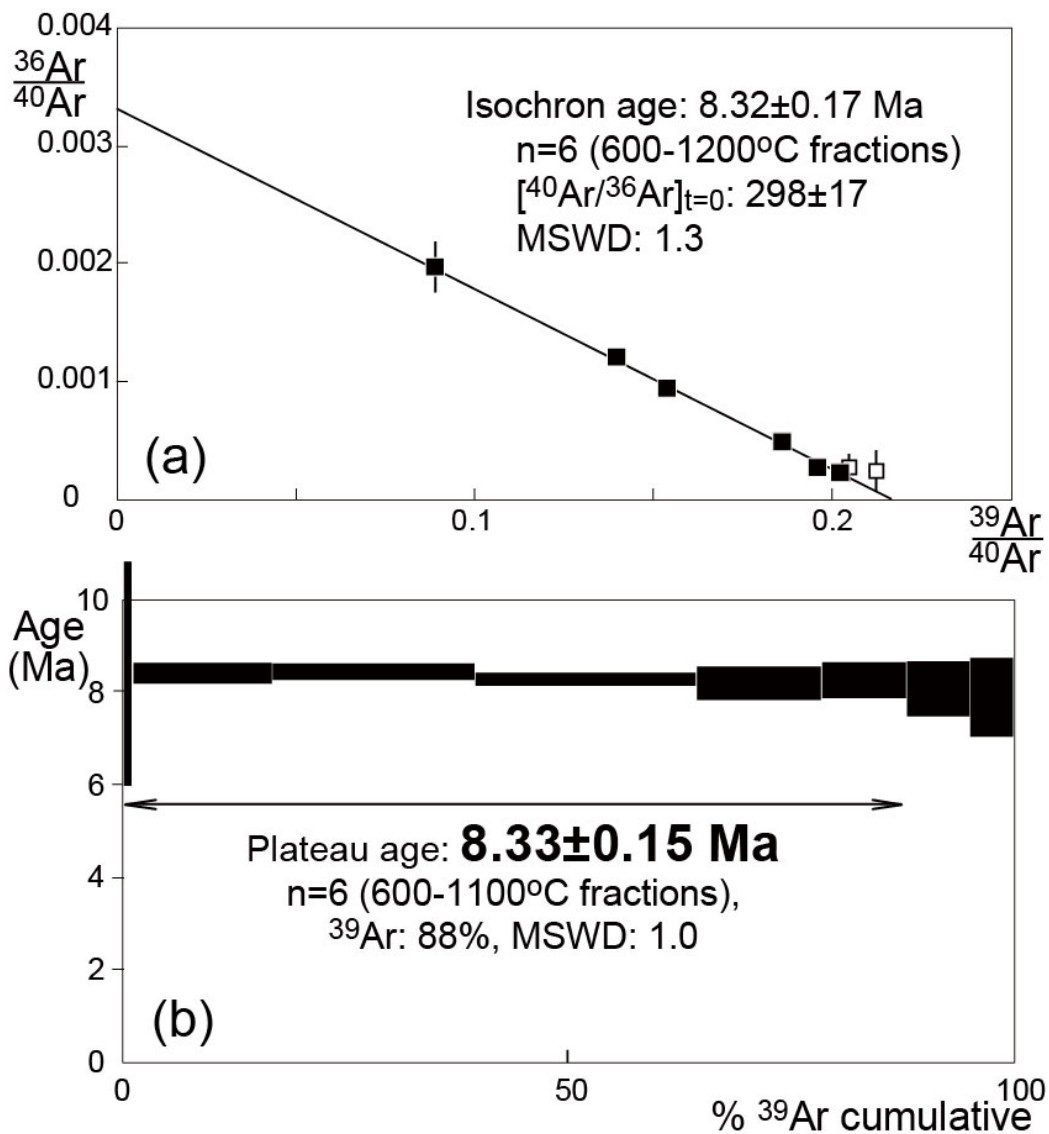

Figure 7: (a) Inverse isochron and (b) $^{40}Ar/^{39}Ar$ age spectrum. MSWD = mean-squared weighted deviation (MSWD = SUMS/(n – 2); York, 1968). All errors are 2σ.

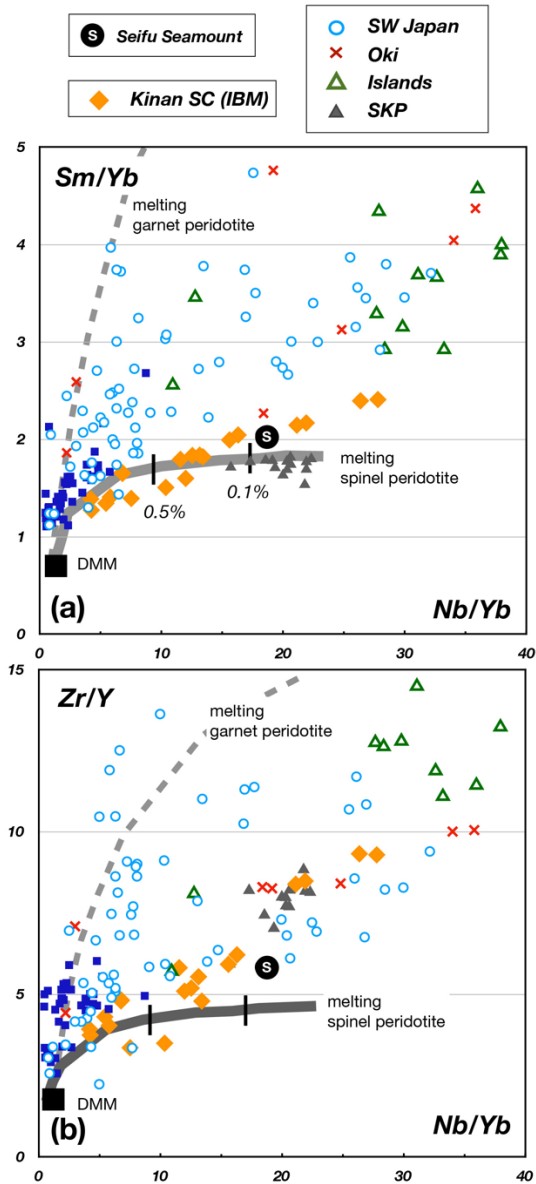

Figure 8: (a) Sm/Yb vs. Nb/Yb. (b) Zr/Y vs. Nb/Yb. Melting trends for garnet and spinel peridotites were calculated from a batch melting model and using the distribution coefficients from Kelemen *et al*. (2003). The depleted MORB mantle (DMM) composition is from Workman and Hart (2005). The southwestern Japan data are from San-yo, San-in, and northwest Kyushu (see Fig. 3 for the data sources).

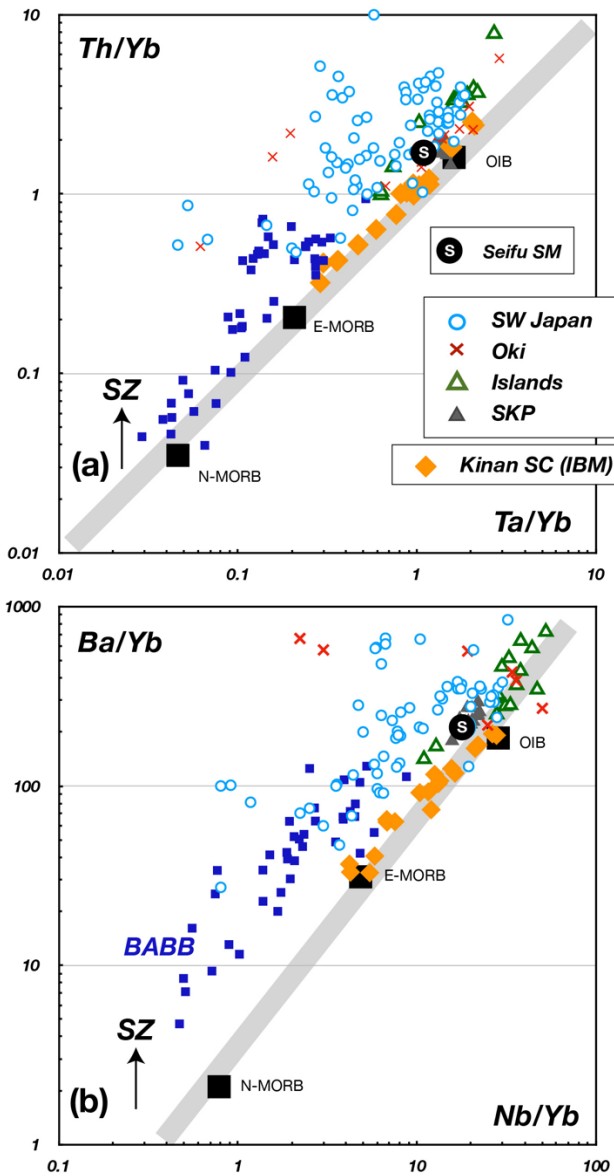

Figure 9: (a) Th/Yb vs. Ta/Yb. (b) Ba/Yb vs. Nb/Yb. The gray line represents the array of global MORB compositions and the arrow indicates the effect of subduction components (SZ) (Pearce and Stern, 2006). Compositions of N-MORB, E-MORB, and OIB are from Sun and McDonough (1989).

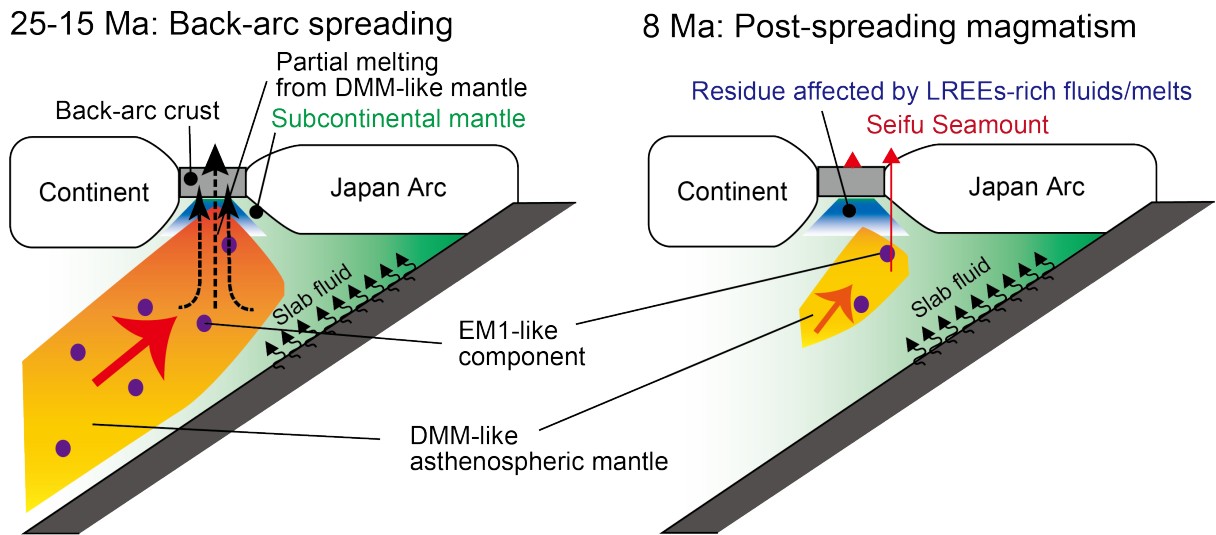

Figure 10: Schematic models showing back-arc spreading magmatism (25-15 Ma) and post-spreading magmatism (8 Ma) in the Japan Sea. The SSM basalt was formed by a low-degree partial melting of spinel peridotite having DMM with a minor EM1-like component during post-spreading magmatism. The SSM basalt contains subcontinental mantle and residue after open system melting affected by infiltration of LREEs-enriched fluids/melts.

