# Peer review of "Alkali basalt from Seifu Seamount in the Japan Sea: post-spreading magmatism in a back-arc setting"

_Solid Earth, 2019_

## Referee Comment (RC1) · Anonymous Referee #1 · 19 Aug 2019

The authors Tomoaki et al. focused the nature of post-backarc magmatism and provided a case study on the Seifu alkali basalt of the Japan Sea, using Ar-Ar geochronology and whole-rock major, trace element and Sr-Nd-Pb isotopic geochemistry. Their methods of direct fusion are interesting and ideal for the analysis of precious samples with a limited amount. The investigated SSM basalt provides a reference for studies on eastern Asian Cenozoic basalts and post-backarc magmatism and plays an important role to link the garnet-pyroxenite/peridotite sourced eastern China Cenozoic basalts and the spinel-peridotite sourced Japan Sea BABB. The primary issue is the organization of the language. The logic relationship of sentences is also weak, especially in the abstract, the 4.2 origin of the SSM basalt and its tectonic setting, and the conclusion part. Luckily, there are not many scientific problems. Thus, I suggest a moderate revision.

Major concerns:

(1) The language is not well organized. Some are not likely English. I rewrote an abstract. But I am also not a native speaker so the new abstract is just for authors' consideration. The main text should be carefully examined for language issues.

Abstract The spreading of the Japan Sea back-arc oceanic floor paused since 18-15 Ma. However, post-backarc magmatism continued. Here we report the geochronological and geochemical results of the Seifu Seamount basalt (SSM-basalt) in the southwest Japan Sea, to reveal its source lithology and heterogeneity. Our results show that the SSM-basalt occurred at 8.33 $\pm$ 0.15 Ma ($2\sigma$) that is an early stage of the Japan Sea post-backarc magmatism, based on the 40Ar-39Ar plateau age. The SSM-basalt is highly alkaline and characterized by enrichment of light rare earth element (LREE) and Nb, similar to ocean island-type basalt (OIB). However, it has an elevated and approximately flat heavy rare earth element (HREE) pattern similar as mid-oceanic ridge basalts (MORB) and back-arc basin basalts (BABB), indicating an origin of low-degree partial melting of spinel peridotite mantle source. The Nd, Sr and Pb isotopic ratios suggest a depleted MORB mantle source that likely added by minor EM1-type enriched components. Here we concluded that the SSM-basalt formed shortly after the main back-arc spreading by low-degree partial melting of spinel peridotite marks a transitional product between the garnet-pyroxenite/peridotite sourced eastern China Cenozoic basalts and the spinel-peridotite sourced Japan Sea BABB.

Keywords Alkali basalt, Japan Sea, Post-backarc magmatism, source lithology, mantle heterogeneity

(2) The discussion 4.2 Origin of the SSM basalt and its tectonic setting is very confusing. Please split it the into two sections, including "4.2 Source lithology and heterogeneity of the SSM basalt" and "4.3 Tectonic origin and implications". Rewrite all of

them. In the new discussion 4.2, please use three paragraphs discuss the magma evolution (was there any AFC processes? How about the Ce/Pb ratio?); source lithology (peridotite/pyroxenite source? using FeO/MnO, CaO/Al2O3 ratios, spinel or garnet facies? La/Yb, Sm/Yb ratios); and source heterogeneity (how to interpret the high Ba/Th ratio?). I am worried that the low 87Sr/86Sr ratio was caused by weathering alteration and the source is isotopically same as the Yamato Basin BABB.

Minor corrections: (1) Rewrite the paragraph 1 in the introduction please. It is boring and not attractive, and not pointed out the scientific issue that the authors are going to solve. The second sentence tells that the post backarc magmatism is characterized by enriched basalts whereas the third sentence tells nothing.

(2) Use "seamounts" instead of the term "seamount chain" in the geological background. The latter makes readers link these seamounts with mantle plumes. However, the authors did not consider any possibility of plume at all in the whole paper.

(3) Please provide thin section photomicrograph images in the main text, showing the olivine, plagioclase, orthopyroxene, clinopyroxene, and spinel phenocrysts.

(4) Direct fusion method is not widely used. Please write the full procedures on major and minor element analysis using EPMA and LA-ICPMS in the main text methods, rather than just say "Details are shown in Tamura et al. (2015).".

(5) Move the geochemical data in the supplementary Table S1 to the main text.

(6) Please provide the original measured EPMA and LA ICP-MS data in the supplementary table.

(7) Revise the "a few mm grains" to be accurate in 3.1. Replace "were done" with "were conducted".

(8) delete the "and high-K to shoshonitic composition subdivided by Le Maitre (1989)" in 3.2. The SSM basalt is obviously sodic.

(9) Please rewrite 3.2 Results. The current version is too short. The elements P, Ti, Al, Ca should be mentioned. Whether it is high-Ti or low-Ti basalt? How about the CaO/Al2O3 and FeO/MnO ratios (pyroxnite/peridotite source lithology)? How about the Ba/Th ratio and its implication? In the discussion timing of the SSM basaltic magmatism is in the first. So please change the order in the methods and results that move the geochronology content in the first.

(10) The conclusion is too long and just repeats the words of the maintext. Shorten to two to three sentences please.
* * *

---

## Referee Comment (RC2) · Anonymous Referee #2 · 22 Aug 2019

This study presents an interesting dataset and has potential for publication but needs significant editing and addition of modelling before acceptance. There are four main problems to address:

(1) The English and structure need to be improved, as commented by reviewer 1.

(2) The dataset is of good quality but limited for a paper. WR geochemical data and an Ar age are provided for only 1 basalt sample. Given inaccessibility of seamounts it is possible that this is a dataset worth considering for publication, but it is currently difficult to understand why this is so. It is very uncommon to publish a paper based on 1 basalt sample, so the introduction needs to explain why the new data is significant

for our understanding of the magmatic/tectonic evolution of the Japan Sea, and what is the key critical question, or set of questions, that the new data allow to address.

(3) The basalt composition is very unusual and interesting. I agree with the authors that there seems (apparently) to be a minor slab-component contribution to the origin of this basalt. However, because of the unusual medium to heavy REE contents (spoon-shaped, not flat, in normalised multielement diagram), I am not convinced that garnet and/or amphibole was not residual in the source. This could be easily addressed by modelling of mantle partial melting.

(4) Finally, it is very surprising that the discussion does not comment on the petrological value of the two types of xenoliths that are abundant in the basalt. The xenoliths were previously studied by Ninomiya et al. 2007, mostly published in Japanese, and only briefly referred to in the beginning of the manuscript. Ninomiya et al. reported the occurrence of (i) mantle xenoliths possibly metasomatically enriched by slab-derived fluids and (ii) xenotliths of subcontinental peridotite. How is it possible then that the studied basalt is coming from a shallow upper mantle that is mostly uncontaminated by continental and slab components? It is critical that this apparent inconsistency is resolved/explained during the revision. Also, could these xenoliths help shed light on the likely processes required to explain the unusual HREE pattern of the studied basalt?

Detailed comments for the authors (mostly to help improve the introduction):

Previous work by Ninomiya et al. 2007 needs to be summarised in the geological background, in particular mentioning the nature of the two types of xenoliths/crysts, because this study (mostly published in Japanese) provides significant constraints and/or context to discuss the source/petrogenesis of the studied carrier basalt. The first mention of the subcontinental nature of some of the xenoliths is currently in the Methods section.

You should not invite the reader to access 6 other papers to understand the Methods -

all methods need to be discussed/summarised in the Methods section.

P.2 L6: What are the 2 distinct magma types referred to here? L7: Define "western Pacific back-arc basin swarm". In addition, this is a very strange term, which I think is unlikey to be understood by many readers. It should be replaced by a more generic term commonly used in international geological literature. L14: "basinS" - Do you actually mean that there are several basins in the back-arc basin, or is this a typo or does it mean there are several oceanic grabens between horsts including continental crust/thinned continental lithosphere? L15-17: What sort of dating? If K-Ar ages, are you confident they represent crystalisation ages? L20: Better to avoid "seamount chain" if there is no hotspot. Perhaps rephrase as "elongated cluster of seamounts". L30: Reference needed for the occurrence of continental crust (seismic imaging evidence?).

P.3 L8: repeat the age of the Yamato seamounts for clarity. L11-19: This looks more like a result section - probably too detailed anyway for an introduction. L15: You should probably tone down "indicating that all orthopyroxene phenocrysts are of xenocryst origin" to "suggest that all...". Alternatively replace "all" by "most".

Figure 1: Is it possible to include a map showing known/generally accepted ages of the oceanic basins (or somehow add this in the current Figure)?

Table S1: What is the error range of TIMS results? Can you provide standard and blank analyses?

---

## Author Comment (AC1) · 13 Oct 2019

Thank you very much for detailed comments. The comments from reviewers improved the manuscript significantly. We hope the revised manuscript would be considered for the publication in Solid Earth.

Please check the attached PDF file and replies below.

1) Primary issue: English Reply: English of the revised version has been improved by a commercial English editing service (but not for this response letter).

2) The discussion 4.2 Origin of the SSM basalt and its tectonic setting. Split it into two

sections: (1) source lithology and heterogeneity of the SSM basalt, and (2) Tectonic origin and implications. Reply: We split subchapter 4.2 into two: (4.2. Origin of the SSM basaltic melt) and 4.3. Tectonic setting of the SSM basalt.

Modeling for melt chemistry: Reply: We modeled calculated melt components between spinel peridotite conditions and garnet peridotite conditions. These results support the SSM melts might be formed at spinel peridotite conditions with low degree of partial melting (New Figure 8) (4-8 lines of page 6). Reply 2: We examined contributions of slab-derived components by correlations between elements of incompatible and less incompatible elements (New Figure 9). Fig. 9 suggest not much slab-derived components were contributed for the formation of the SSM melt (9-13 lines of page 6).

Effects of weathering on Sr-isotopes: Reply: We showed leached and unleached data and these data supported that isotopic compositions have not been significantly affected by the alteration. (8-10 lines of page 5)

Minor (1) Rearrangement of introduction Reply: As we also reply to a question from reviewer 2, we also includes why only one basaltic sample (because this sample only contains mantle-derived xenolith)(9-14 lines of page 2) was examined in this paper in introduction. The motivation and results might not be a "Wow" type paper (I am sure that it is very "Wow" for the authors), but we strongly believe that our results contribute to know back-arc magmatism, and our introduction leads the readers to be interested in back-arc magmatic history.

(2) Use "seamounts" instead of the term "seamount chain" in the geological background. The latter makes readers link these seamounts with mantle plumes. However, the authors did not consider any possibility of plume at all in the whole paper. Reply: We changes seamount chain to seamounts. But Yamato Seamount Chain and Kinan Seamount Chain have been used in previous papers. We use Seamount Chain (S and C are uppercase) for these seamounts.

(3) provide thin section image showing olivine, plagioclase, opx, cpx and spinel phenocrysts. Reply: We prepared two images (both polarized light and cross polarized images) for a tiny xenolith-bearing one and matrix (New Figure 2). These doe not contain cpx, opx, and spinel phenocryst, but the readers can see the typical texture of the SSM basalt and can understand why we assume that some opx and spinel phenocrysts were of xenocryst origin.

(4) direct fusion method Reply: We described a brief summary of the method of direct fusion (24-28lines of page 3)

(5, 6) supplementary table S1 to the main text, and original EPMA-LAICPMS data in the text Reply: We did.

(7) Revise the "a few mm grains" to be accurate in 3.1. Replace "were done" with "were conducted". Reply: We did (< 2mm and conducted, respectively).

(8) delete the "and high-K to shoshonitic composition subdivided by Le Maitre (1989)" in 3.2. The SSM basalt is obviously sodic. Reply: We deleted this sentence.

(9) Rewrite 3.2. Results. Reply: We added several lines of information, such as low-Ti basalt, MgO, and FeO/MgO ratio. FeO/MnO ratio was used for discussions.

(10) Conclusion is too long: Reply: The conclusion was shortened to focus on the most important results of this study (24-27 lines of page 7).

Please also note the supplement to this comment:
https://www.solid-earth-discuss.net/se-2019-116/se-2019-116-AC1-supplement.pdf

**Supplement:**

[revised manuscript text omitted]
*., 2015). The contribution of slab-derived components to the mantle source of the SSM basalt was minor. Peridotite xenoliths in the SSM basalt are not directly related to its petrogenesis. The SSM basalt was likely a product of heterogeneous asthenospheric mantle as discussed in the next section.

**4.3. Tectonic setting of the SSM basalt**

The age and geochemistry of the SSM basalt suggest that upwelling of the Japan Sea back-arc asthenosphere continued after the cessation of back-arc spreading, such that low-degree partial melting could occur. The Nd, Sr, and Pb isotopic compositions of the SSM basalt are also clearly different from those of alkali basalts in the circum-Japan Sea area. In Nd–Sr and Pb isotope diagrams, data for the SSM basalt is slightly offset from the compositional trend of Japan Sea BABB magmatism (Fig. 7). 
[revised manuscript text omitted]

---

## Author Comment (AC2) · 13 Oct 2019

Thanks very much for the detailed comments. The comments from reviewers improved the manuscript significantly. We hope the revised manuscript would be considered for publication in Solid Earth.

English Reply: English of the revised version has been improved by a commercial English editing service (but not for this response letter).

(1) Why only one sample? Reply: This is the only sample contains the mantle-derived sample. We described the importance of the description of this sample in "Introduction"

(9-14 lines of page 2).

(2) Modeling for REE pattern. Reply: We modeled calculated melt components between spinel peridotite conditions and garnet peridotite conditions. These results support that the SSM melts might be formed at spinel peridotite conditions with a low degree of partial melting (New Figure 8) (4-8 lines of page 6).

(4) Xenolith information Reply: We briefly summarized xenolith reported by Ninomiya et al. (2017) (5-9 lines of page 3). Our data suggest that peridotite xenoliths in the sample are not directly related to peterogenesis of the SSM basalt (21 line page 6).

Detailed comments. Others: Analytical methods should be described. Reply: We added the minimum necessary description of analytical methods (See analytical methods).

Other comments are related to the introduction Reply: We rearrange and changed introduction as suggested above including several comments from reviewer 2.

Age data in Figure 1: Reply: We added age data in the revised Figure 1.

Table S1 (Now Table 1) Error range of TIMS results Reply: We showed the error of the 2sigma range.

Please also note the supplement to this comment:
https://www.solid-earth-discuss.net/se-2019-116/se-2019-116-AC2-supplement.pdf

**Supplement:**

[revised manuscript text omitted]
*., 2015). The contribution of slab-derived components to the mantle source of the SSM basalt was minor. Peridotite xenoliths in the SSM basalt are not directly related to its petrogenesis. The SSM basalt was likely a product of heterogeneous asthenospheric mantle as discussed in the next section.

**4.3. Tectonic setting of the SSM basalt**

The age and geochemistry of the SSM basalt suggest that upwelling of the Japan Sea back-arc asthenosphere continued after the cessation of back-arc spreading, such that low-degree partial melting could occur. The Nd, Sr, and Pb isotopic compositions of the SSM basalt are also clearly different from those of alkali basalts in the circum-Japan Sea area. In Nd–Sr and Pb isotope diagrams, data for the SSM basalt is slightly offset from the compositional trend of Japan Sea BABB magmatism (Fig. 7). 
[revised manuscript text omitted]

---

## Author Response (AR2)

Thanks for the editor's 3 comments that have greatly improved the manuscript.

Here we explained how we corrected the manuscript point by point.

New sentences are shown in blue, and rearranged sentences (the sentence is the same as the previous manuscript but is changed to the position) are in green in the revised manuscript.

Comment1: how is it possible that a basalt that originates from a relatively shallow DMM-EM1 source carries two distinct types of mantle xenoliths, including one derived from subcontinental lithospheric mantle? What does this imply about the tectonic setting and evolution? In response, you write that 'Peridotite xenoliths in the SSM basalt are not directly related to its petrogenesis'; this may be so, but it does not provide an answer to the question that the reviewer has raised. This is especially pertinent given that you state in the (revised) introduction that one of the aims is 'to evaluate the significance of these xenoliths'.

Reply: We rearranged several sentences to discuss how mantle xenoliths were captured in the SSM basalt (13-19 lines of page 7). We prepared a new figure to support the idea (Fig. 10). Then we explained how two types of mantle xenoliths were captured in the SSM basalt as follows: "*Based on our study of the SSM basalt coupled with earlier preliminarily works on peridotite xenoliths in the SSM basalt (Ninomiya et al., 2007) suggest peridotite xenoliths in the SSM basalt seem to be not directly related to the SSM basalt petrogenesis (Fig. 10). One of the two types of peridotite xenoliths is interpreted as fragments of subcontinental lithospheric mantle that were already located before the Japan Sea opening. The other type, which is characterized by residue after open system melting caused by infiltration of LREEs-enriched fluids, may be related to back-arc spreading magmatisms affected by slab-derived fluids/melts (Fig. 10). Further study of peridotite xenoliths and mafic xenoliths is needed to reconstruct the crust-mantle evolution beneath the Japan Sea*".

[Figure]

Figure 10: Schematic models showing back-arc spreading magmatism (25-15 Ma) and post-spreading magmatism (8 Ma) in the Japan Sea. The SSM basalt was formed by a low-degree partial melting of spinel peridotite having DMM with a minor EM1-like component during post-spreading magmatism. The SSM basalt contains subcontinental mantle and residue after open system melting affected by infiltration of LREEs-enriched fluids.

Comment 2: In response to Reviewer 1's worry that the isotopic data may be compromised by alteration, I welcome the addition of the data on leached vs unleached samples. You comment that 'although these isotopic compositions are slightly different, alteration has not significantly affected the isotopic data'. Please elaborate on this statement: it would appear that the Sr isotopes, in particular, show offsets between the leached and unleached samples that are consistent with an alteration contribution to the unleached measurements.

Reply: We corrected figure (Figure 6a) including unleached and leached data. The figure clearly shows the slight difference between samples but not significant.

Comments3: it would appear that the Ar/Ar data are not reported: could you please add a table with these data?

Reply: We prepared new Table 3 showing Ar/Ar data.

Thanks indeed for supporting our manuscript with valuable comments.

Addition to this, a new affiliation, JAMSTEC, is added to Tomoaki MORISHITA.

Best regards,

Tomo MORISHITA, corresponding author

---

## Author Response (AR3)

14 Nov, 2019

Dear Johan,

Thanks indeed for your careful handling, supports and encouragements. We corrected Figure 10 as you suggested.

Best regards,

Tomo MORISHITA, corresponding author.